# Genome-wide meta-analysis identifies five new susceptibility loci for pancreatic cancer

Alison P. Klein *et al.*[#]

In 2020, 146,063 deaths due to pancreatic cancer are estimated to occur in Europe and the United States combined. To identify common susceptibility alleles, we performed the largest pancreatic cancer GWAS to date, including 9040 patients and 12,496 controls of European ancestry from the Pancreatic Cancer Cohort Consortium (PanScan) and the Pancreatic Cancer Case-Control Consortium (PanC4). Here, we find significant evidence of a novel association at rs78417682 (7p12/*TNS3*, $P = 4.35 \times 10^{-8}$). Replication of 10 promising signals in up to 2737 patients and 4752 controls from the PANcreatic Disease ReseArch (PAN-DoRA) consortium yields new genome-wide significant loci: rs13303010 at 1p36.33 (*NOC2L*, $P = 8.36 \times 10^{-14}$), rs2941471 at 8q21.11 (*HNF4G*, $P = 6.60 \times 10^{-10}$), rs4795218 at 17q12 (*HNF1B*, $P = 1.32 \times 10^{-8}$), and rs1517037 at 18q21.32 (*GRP*, $P = 3.28 \times 10^{-8}$). rs78417682 is not statistically significantly associated with pancreatic cancer in PANDoRA. Expression quantitative trait locus analysis in three independent pancreatic data sets provides molecular support of *NOC2L* as a pancreatic cancer susceptibility gene.

#A full list of authors and their affliations appears at the end of the paper.

Pancreatic cancer is currently the third leading cause of cancer-related deaths in the United States and the fifth leading cause in Europe[1,2], and is predicted to become the second leading cause of cancer-related deaths in the United States by 2030[3,4]. Incidence rates of pancreatic cancer have also gradually increased[1]. Genetic susceptibility plays an important role in pancreatic cancer risk through mutations in known genes for hereditary cancer or hereditary pancreatitis[5–11], and common genetic variants identified through genome-wide association studies (GWAS)[12–16].

With the aim of identifying additional common pancreatic cancer risk loci, the Pancreatic Cancer Cohort Consortium (PanScan: https://epi.grants.cancer.gov/PanScan/) and the Pancreatic Cancer Case-Control Consortium (PanC4: http://www.panc4.org/) have performed GWAS of pancreatic ductal adenocarcinoma (PDAC) in populations of European ancestry. These studies, namely PanScan I[12], PanScan II[13], PanScan III[14,15], and PanC4[16], have led to the identification of 13 genomic loci carrying 17 independent pancreatic cancer risk signals on chromosomes 1q32.1 (two independent signals in *NR5A2*), 2p14 (*ETAA1*), 3q28 (*TP63*), 5p15.33 (three independent risk loci in the *CLPTM1L-TERT* gene region), 7p14.1 (*SUGCT*), 7q23.2 (*LINC-PINT*), 8q24.1 (two independent risk loci in the *MYC-PVT1* gene region), 9q34.2 (*ABO*), 13q12.2 (*PDX1*), 13q22.1 (non-genic), 16q23.1 (*BCAR1*), 17q24.3 (*LINC00673*), and 22q12.1 (*ZNRF3*)[12–16]. A fourth independent risk locus at 5p15.33 (*TERT*) was identified through a candidate gene analysis by the PANcreatic Disease ReseArch (PANDoRA) case–control consortium[17,18]. GWAS in populations from China[19] and Japan[20] have identified eight additional GWAS significant pancreatic cancer risk loci on chromosomes 5p13.1 (*DAB2*), 6p25.3 (*FOXQ1*), 7q36.2 (*DPP6*), 12p11.21 (*BICD1*), 10q26.11 (*PRLHR*), 21q21.3 (*BACH1*), 21q22.3 (*TFF1*), and 22q13.32 (*FAM19A5*)[19]. The overlap among loci identified in the European and Asian ancestry scans with current sample sizes is limited[14,16,21].

Here we report the findings of the largest Pancreatic Cancer GWAS study to date. Five novel regions of association were identified. A locus at rs78417682 (7p12/*TNS3*, $P = 4.35 \times 10^{-8}$) was identified in meta-analysis of the PanScanI/II, PanScanIII, and PanC4 data. Four additional loci, rs13303010 at 1p36.33 (*NOC2L*, $P = 8.36 \times 10^{-14}$), rs2941471 at 8q21.11 (*HNF4G*, $P = 6.60 \times 10^{-10}$), rs4795218 at 17q12 (*HNF1B*, $P = 1.32 \times 10^{-8}$), and rs1517037 at 18q21.32 (*GRP*, $P = 3.28 \times 10^{-8}$) were identified after replication in additional cases and controls from the PANDoRA consortium.

## Results

**Association analysis**. In the current study, we performed the largest association analysis of pancreatic cancer risk to date, including 9040 individuals diagnosed with pancreatic cancer and 12,496 control individuals of European ancestry (Supplementary Table 1) from four GWAS studies (PanScan I, PanScan II, PanScan III, and PanC4). These individuals were previously genotyped on the Illumina HumanHap550, 610-Quad, OmniExpress, and OmiExpressExome arrays, respectively[12–16]. Because of the extensive overlap of variants on the arrays, data from PanScan I and PanScan II were analyzed jointly, while PanScan III and PanC4 were each analyzed separately. Imputation was performed using the 1000 G (Phase 3, v1) reference data set[22]. After quality control 11,381,182 variants were analyzed for 21,536 individuals (7167 in PanScan I+II, 6785 in PanScan III, and 7584 in PanC4). A quantile–quantile plot (Supplementary Figure 1) showed little evidence of systematic inflation ($\lambda = 1.002$ for PanScan I+II, $\lambda = 1.051$ for PanScan III, $\lambda = 1.025$ for PanC4, and $\lambda = 1.05$ for the meta-analysis).

In a fixed-effect meta-analysis of PanScan I+II, PanScan III, and PanC4, we observed robust associations at our previously identified susceptibility loci in individuals of European ancestry[12–16] (Supplementary Table 2). We also noted one novel locus that met the genome-wide significance threshold ($P < 5 \times 10^{-8}$: Wald test) at chromosome 7p12 in the *TNS3* gene, and nine additional promising loci ($P < 1 \times 10^{-6}$: Wald test). These 10 loci were carried forward to an independent replication in up to 2737 pancreatic cancer cases and 4752 control individuals from the PANcreatic Disease ReseArch (PANDoRA) consortium[23]. In a combined meta-analysis of up to 11,537 pancreatic cancer cases and 17,107 control individuals from PanScan I+II, PanScan III, PanC4, and PANDoRA, we identified three additional loci of genome-wide significance: rs13303010 at 1p36.33 (*NOC2L*, odds ratio (OR) = 1.26; 95% confidence interval (CI) 1.19–1.35, $P = 8.36 \times 10^{-14}$: Wald test), rs2941471 at 8q21.11 (*HNF4G*, OR = 0.89, 95% CI 0.85–0.93, $P = 6.60 \times 10^{-10}$: Wald test), and rs4795218 at 17q12 (*HNF1B*, OR = 0.88, 95% CI 0.84–0.92, $P = 1.32 \times 10^{-8}$: Wald test). A locus that was previously reported to be suggestive in the PanC4 study at 18q21.32 in the *GRP* gene[16] also surpassed the significance threshold (rs1517037, OR = 0.86, 95% CI 0.80–0.91, $P = 3.28 \times 10^{-8}$: Wald test) in our meta-analysis. (Table 1 and Fig. 1). The single-nucleotide polymorphism (SNP) at 7p12 in *TNS3* (rs73328514) was not significantly associated with pancreatic cancer in PANDoRA (OR = 0.94, $P_{\text{PANDoRA}} = 0.31$; OR = 0.85, $P_{\text{Combined}} = 1.35 \times 10^{-7}$: Wald test).

The marker SNP at 1p36.33 (rs13303010) maps to the first intron of the *NOC2L* gene, which encodes the nucleolar complex protein 2 homolog (NOC2-like protein, also known as novel INHAT repressor), an inhibitor of histone acetyltransferase (HAT) activity and transcriptional repressor[24]. This protein also directly binds to p53, stabilizing an interaction between the mitotic kinase Aurora B and p53, resulting in inhibition of p53-mediated transcriptional activation[24,25]. Likewise, NOC2-like protein binds and inhibits transcriptional activity of the closely related tumor suppressor protein, p63 (TAp63)[26]. Notably, we have previously identified a pancreatic cancer risk locus intronic to the *TP63* gene[16].

At chromosome 8q21.11, the newly associated SNP (rs2941471) is intronic to *HNF4G*, which encodes hepatocyte nuclear factor 4 gamma, a transcription factor (TF) of the nuclear receptor superfamily. Mice lacking *Hnf4g* have higher numbers of pancreatic β-cells, increased glucose-induced insulin secretion, and improved glucose tolerance[27]. Of the multiple GWAS that have reported association signals at this locus for other conditions, the variant with the highest linkage disequilibrium (LD) with our pancreatic cancer-associated variant, rs2941471, has been significantly associated with variations in serum urate concentrations (rs2941484[28], $r^2 = 0.56$, in 1000 G EUR). Interestingly, we have previously shown that urate levels are associated with pancreatic cancer risk[29]. Additional variants in the *HNF4G* gene region, including those significantly associated with body mass index (BMI)[30], obesity[31], and breast cancer[32] are less correlated with rs2941471 ($r^2 = 0.02$–0.12).

The signal at 17q12 (rs4795218) maps to the fourth intron of *HNF1B*, encoding another member of the hepatocyte nuclear factor family. HNF1B plays an important role in pancreatic development, acting in a transcriptional network that controls the differentiation of multipotent progenitor cells to acinar, ductal, and endocrine cells[33,34]. Mutations in *HNF1B* account for a small percentage (1–2%) of maturity onset diabetes of the young (MODY)[35]. In addition, variants in the *HNF1B* gene region that are modestly linked with rs4795218 have previously been associated with the development of prostate cancer (rs4794758, $r^2 = 0.59$ in 1000 G EUR)[36]. Although additional variants in this

**Table 1 Novel pancreatic cancer susceptibility loci**

| Chr[a] SNP Position[b] gene | Effect allele (minor)/ reference allele | Statistic | PanScan I/II 3535 cases and 3642 controls | PanScan III 1582 cases and 5203 controls | PanC4 3933 cases and 3651 controls | ALL GWAS 9040 cases and 12,496 controls | PANDoRA 2497 cases and 4611 controls | GWAS +PANDoRa 11,537 cases and 17,107 controls |
|---|---|---|---|---|---|---|---|---|
| 1p36.33 rs13303010 894,573 NOC2L | G/A | MAF[c] cases; controls | 0.14; 0.13 | 0.12; 0.10 | 0.13; 0.11 | | 0.14; 0.10 | – |
| | | Info[d] | 0.42 | g | g | | g | – |
| | | OR (CI) | 1.15 (1.01–1.26) | 1.22 (1.09–1.33) | 1.16 (1.07–1.24) | 1.20 (1.12–1.29) | 1.45 (1.33–1.57) | 1.26 (1.19–1.35) |
| | | $P$ value | $3.64 \times 10^{-2}$ | $1.48 \times 10^{-3}$ | $9.54 \times 10^{-4}$ | $7.30 \times 10^{-7}$ | $6.00 \times 10^{-10}$ | $8.36 \times 10^{-14}$ |
| | | Heterogeneity $P$ value[e] | | | | $6.49 \times 10^{-1}$ | | $4.57 \times 10^{-2}$ |
| 7p12 rs73,328,514 47488569 TNS3 | T/A | MAF cases; controls | 0.09; 0.11 | 0.10; 0.12 | 0.10; 0.12 | | 0.10; 0.11 | – |
| | | Info | 0.93 | 0.97 | 0.97 | | g | – |
| | | OR (CI) | 0.80 (0.71–0.89) | 0.88 (0.76–1.02) | 0.82 (0.74–0.92) | 0.83 (0.77–0.88) | 0.94 (0.83–1.06) | 0.85 (0.80–0.90) |
| | | $P$ value | $8.38 \times 10^{-5}$ | $9.31 \times 10^{-2}$ | $3.61 \times 10^{-4}$ | $4.35 \times 10^{-8}$ | $3.08 \times 10^{-1}$ | $1.35 \times 10^{-7}$ |
| | | Heterogeneity $P$ value | | | | $5.98 \times 10^{-1}$ | | $2.35 \times 10^{-1}$ |
| 8q21.11 rs2941471 76,470,404 HNF4G | G/A | MAF cases; controls | 0.40; 0.43 | 0.41; 0.42 | 0.41; 0.43 | | 0.40; 0.43 | |
| | | Info | 1.0 | 1.0 | 1.0 | | g | |
| | | OR (CI) | 0.87 (0.79–0.94) | 0.91 (0.80–1.01) | 0.89 (0.82–0.96) | 0.89 (0.86–0.94) | 0.87 (0.79–0.94) | 0.89 (0.85–0.93) |
| | | $P$ value | $2.39 \times 10^{-4}$ | $8.30 \times 10^{-2}$ | $2.19 \times 10^{-3}$ | $4.73 \times 10^{-7}$ | $2.42 \times 10^{-4}$ | $6.60 \times 10^{-10}$ |
| | | Heterogeneity $P$ value | | | | $7.73 \times 10^{-1}$ | | $7.87 \times 10^{-1}$ |
| 17q12 rs4795218 36,078,510 HNF1B | A/G | MAF cases; controls | 0.20; 0.23 | 0.22; 0.23 | 0.21; 0.23 | | 0.21; 0.23 | |
| | | Info | 0.96 | 0.96 | 0.95 | | g | |
| | | OR (CI) | 0.87 (0.80–0.95) | 0.88 (0.78–0.98) | 0.88 (0.81–0.95) | 0.88 (0.82–0.93) | 0.90 (0.82–0.98) | 0.88 (0.84–0.92) |
| | | $P$ value | $1.12 \times 10^{-3}$ | $2.29 \times 10^{-2}$ | $1.11 \times 10^{-3}$ | $2.73 \times 10^{-7}$ | $1.38 \times 10^{-2}$ | $1.32 \times 10^{-8}$ |
| | | Heterogeneity $P$ value | | | | $9.96 \times 10^{-1}$ | | $9.78 \times 10^{-1}$ |
| 18q21.32 rs1517037 56,878,274 *GRP* | T/C | MAF cases; controls | 0.16; 0.19 | 0.17; 0.19 | 0.17; 0.18 | | 0.17; 0.19 | |
| | | Info | g | g | g | | g | – |
| | | OR (CI) | 0.82 (0.75–0.89) | 0.92 (0.82–1.04) | 0.90 (0.83–0.98) | 0.87 (0.82–0.93) | 0.87 (0.79–0.97) | 0.86 (0.80–0.91) |
| | | $P$ value | $7.56 \times 10^{-6}$ | $1.90 \times 10^{-1}$ | $1.64 \times 10^{-2}$ | $8.81 \times 10^{-7}$ | $1.17 \times 10^{-2}$ | $3.28 \times 10^{-8}$ |
| | | Heterogeneity $P$ value | | | | $1.87 \times 10^{-1}$ | $7.73 \times 10^{-2}$ | $1.03 \times 10^{-1}$ |

[a] Cytogenetic regions according to NCBI Human Genome Build 37
[b] SNP position according to NCBI Human Genome Build 37
[c] Minor allele frequency
[d] Quality of imputation metric. See online methods for more detail. If a SNP is genotyped and not imputed, a "g" is reported
[e] $P$ value from test of heterogeneity

region have been associated with other cancers including prostate[36], endometrial[37], and testicular[38] cancers, they do not appear to mark the same signal (rs4430796, rs11263763, rs7501939, respectively, $r^2$ with rs4795218 ≤ 0.005 in 1000 G EUR).

The two novel risk loci in genes of the hepatocyte nuclear factor family are intriguing in light of our previously published suggestive evidence of association with other members of this family, including a locus at 12q24.31 in the *HNF1A* gene (rs1182933, OR = 1.11, $P = 3.49 \times 10^{-7}$: Wald test) and a locus on 20q13.11 ~20 kb downstream of the *HNF4A* gene (rs6073450, OR = 1.09, $P = 4.55 \times 10^{-6}$: Wald test; Supplementary Table 3)[16]. Members of this family of TFs play important roles in pancreatic development and regulate specific gene expression programs in pancreatic acini, pancreatic islets, and hepatocytes in adults[39–42]. Importantly, HNF1A appears to be a critical member of a signaling network that maintains homeostasis in the adult pancreas[40,43]. We have also previously shown that *HNF1A* may be a tumor suppressor gene in the pancreas[44,45]. Inherited mutations in several genes of this family cause pancreatic beta cell dysfunction resulting in MODY: HNF4A (MODY 1), HNF1A (MODY 3), and HNF1B (MODY 5)[35]. Common variants in or close to some of these genes have also been significantly associated with type 2 diabetes (T2D) and body mass index (BMI)/obesity, both known epidemiologic risk factors for pancreatic cancer[46]. However, the low LD between those signals on 8q21.11/*HNF4G* (PDAC-rs2941471 and BMI-rs17405819[47], $r^2 = 0.05$), 17q12/*HNF1B* (PDAC-rs4795218 and T2D-rs4430796[48], $r^2 = 0.0007$), and 12q24.31/*HNF1A* (PDAC-rs7310409 and T2D-rs12427353[48], $r^2 = 0.18$) indicates that the underlying functional mechanism for the pancreatic cancer GWAS signals may differ from those for adult-onset T2D and BMI.

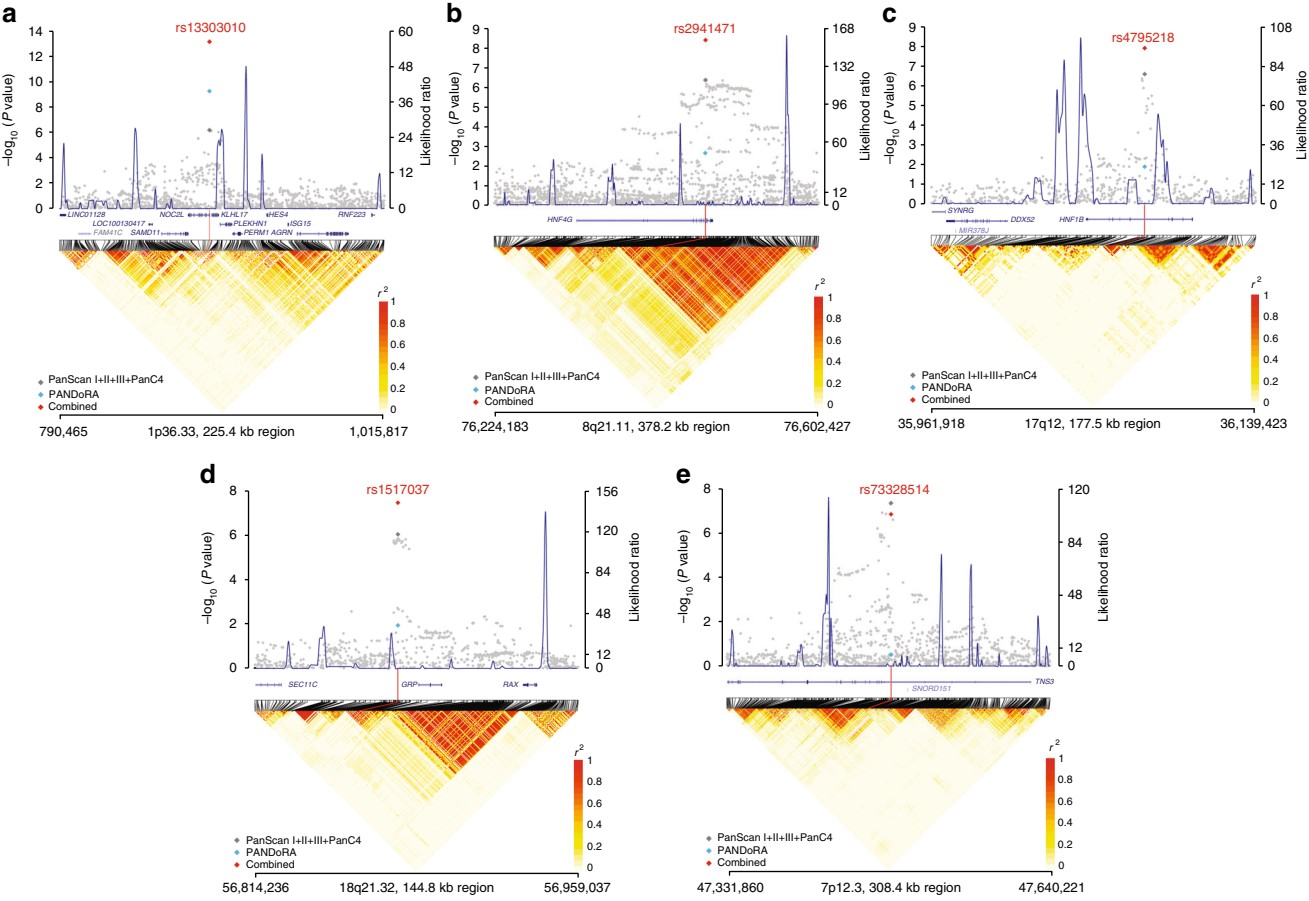

**Fig. 1** Association results, recombination hotspots, and LD plots for new pancreatic cancer susceptibility regions. The top half of each panel shows the association results for the meta-analysis of PanScan I+II, PanScan III, and PanC4 (gray diamonds). The results for the replication of the marker SNP at each locus are shown for PANDoRA (light blue diamonds) and the combined meta-analysis results (red diamonds). Overlaid are likelihood ratio statistics estimating putative recombination hotspots across the region based on the inference using the CEU 1000 G Phase 3 data. Genomic coordinates are plotted on the x axis (Genome build hg19), P values for the association analysis are shown on the left y axis, and recombination hotspot likelihood ratio on the right y axis. The bottom half of each panel shows LD heat maps based on $r^2$ values from the 1000 G Phase 3 CEU population for all variants included in the analysis. Shown are results for chromosomes 1p36.33 (**a**), 8q21.11 (**b**), 17q12 (**c**), 18q21.32 (**d**), and 7p12 (**e**)

At 18q21.32, the signal marked by rs1517037 is ~10 kb upstream of the *GRP* gene, which encodes a member of the bombesin-like family of gastrin-releasing peptides that stimulates the release of gastrointestinal hormones, including amylase[49], a marker of acute pancreatitis. Correlated variants at this locus ($r^2$ = 0.82–1.00) are associated with inflammatory bowel disease (IBD)[50] and BMI[47].

The locus at 7p12 is marked by an intronic SNP (rs73328514) in *TNS3*. This gene encodes Tensin-3, a member of a family of focal adhesion-associated proteins (Tensin-1 through Tensin-4) that regulate cell adhesion and migration[51] and may play a role in metastasis[52].

We identified suggestive evidence for additional risk loci in the meta-analysis of PanScan and PanC4 data; at 9q31.1 in the *SMC2* gene (rs2417487, $P = 1.49 \times 10^{-7}$: Wald test), at 4q31.22 near *EDNRA* (rs6537481, $P = 1.15 \times 10^{-7}$: Wald test), and at 16q24.1 near *LINC01081/LINC00917* (rs7200646, $P = 1.39 \times 10^{-7}$: Wald test) but these were not significantly associated in PANDoRA ($P$ = 0.38, 0.91 and 0.93, respectively: Wald tests; Supplementary Table 4). *SMC2* encodes a subunit of condensin and is necessary for chromosome organization, cell division, and DNA repair[53]. *EDNRA* encodes the endothelin-1 receptor and has been associated with pancreatic cancer prognosis[54]. The locus on 16q24.1 lies ~200–300 kb upstream of a cluster of genes of the forkhead family of TFs (*FOXF1*, *FOXC2*, and *FOXL1*), known for their roles in development, cell proliferation, and several diseases, including cancer[55].

We further estimated a polygenetic risk score (PRS) for pancreatic cancer using the 22 independent genome-wide significant risk SNPs in the Caucasian population[12–16]. The OR for pancreatic cancer among individuals above the 90th percentile the risk distribution was 2.20 (95% CI 1.83–2.65) compared with those with a PRS in the 40–60th percentile (Supplementary Table 5). We also assessed eight pancreatic cancer risk loci identified in Chinese and Japanese individuals in our data and noted one nominally significant locus in the combined PanScan and PanC4 results (6p25.3, rs9502893, OR = 0.94, 95% CI 0.92–0.97, $P = 0.009$: Wald test; Supplementary Table 6).

**Pathway enrichment analysis.** Pathway enrichment analysis for genes in currently known pancreatic cancer risk loci was performed using gene set enrichment analysis (GeneCodis; http://genecodis.cnb.csic.es/analysis)[56] and Data-Driven Expression Prioritized Integration for Complex Traits (DEPICT; https://data.broadinstitute.org/mpg/depict/)[57]. The most significant enrichment was seen for the terms "Maturity onset diabetes of the young" (Kyoto Encyclopedia of Genes and Genomes (KEGG), $P = 5.5 \times 10^{-9}$, Hypergeometric distribution test), "Sequence-specific DNA-binding transcription factor activity" (GO Molecular

**Table 2 Expression quantitative trait loci (eQTLs) for marker SNPs on chromosomes 1p36.33 and 8q21.11 in histologically normal pancreatic tissue samples from GTeX (n = 149) and LTG (n = 95) as well as pancreatic tumor samples from TCGA (PDAC, n = 115)**

**Chr1p36.33: eQTLs for rs13303010**

| Gene name | GTeX pancreas | | LTG pancreas | | TCGA PDAC | |
|---|---|---|---|---|---|---|
| | P value | Effect size* | P value | Effect size* | P value | Effect size* |
| KLHL17 | $2.10 \times 10^{-5}$ | −0.42 | 0.131 | −0.32 | 0.654 | −0.11 |
| NOC2L | 0.001 | 0.39 | 0.019 | 0.41 | 0.043 | 0.49 |
| SAMD11 | 0.023 | −0.26 | 0.500 | 0.14 | 0.397 | −0.18 |
| DVL1 | 0.042 | −0.14 | 0.280 | 0.18 | 0.085 | −0.37 |

**Chr8q21.11: eQTLs for rs2941471**

| Gene name | GTeX pancreas | | LTG pancreas | | TCGA PDAC | |
|---|---|---|---|---|---|---|
| | P value | Effect size* | P value | Effect size* | P value | Effect size* |
| HNF4G | 0.038 | 0.15 | 0.024 | 0.28 | 0.803 | −0.029 |

Expression QTLs were assessed in GTeX pancreatic tissue samples for all RefSeq genes within a 1MB region centered on the marker SNP at each locus. Nominally significant findings were attempted for replication in the LTG and TCGA pancreatic tissue samples. *Effect size is the estimated eQTL effect size or beta ($\beta$) and its direction is shown for the risk increasing allele at each locus

Function, $P = 3.1 \times 10^{-4}$), "Cellular response to UV" (GO Biological Process, $P = 4.2 \times 10^{-4}$, Hypergeometric distribution test) as well as multiple gastrointestinal tissues (DEPICT, $P = 5.1 \times 10^{-5}$ −0.004, Welch's t-test; Supplementary Tables 7 and 8).

**Expression analysis.** To begin unraveling the functional consequences of the newly discovered risk alleles, we performed expression quantitative trait locus (eQTL) analyses in three independent pancreatic tissue sample sets. We first assessed eQTLs in the publicly available Genotype-Tissue Expression (GTeX) project data for 149 histologically normal pancreatic tissue samples (including genes in a 1MB window centered on the marker SNP at each locus). Nominally significant eQTLs ($P < 0.05$) from this analysis (Supplementary Table 9, Supplementary Figure 2) were then carried forward to replication in two additional sample sets: (1) 95 histologically normal pancreatic samples (Laboratory of Translational Genomics, Laboratory of Translational Genomics (LTG) set[58]) and (2) 115 pancreatic tumors (The Cancer Genome Atlas, TCGA, Pancreatic Adenocarcinoma, PAAD, samples[58]; Table 2). The most notable eQTL in this analysis was seen for 1p36.33, where the risk-increasing allele at rs13303010 was associated with higher NOC2L expression in all three data sets (GTeX: $P = 0.01$, $\beta = 0.39$; LTG: $P = 0.019$, $\beta = 0.41$; TCGA: $P = 0.043$, $\beta = 0.49$: T-statistic; Fig. 2b). An additional eQTL for a nearby gene, KLHL17, was significant in GTeX ($P = 2.1 \times 10^{-5}$, $\beta = -0.42$: T-statistic) but not in LTG ($P = 0.131$, $\beta = -0.32$: T-statistic) or TCGA ($P = 0.654$, $\beta = -0.11$: T-statistic). At 8q21.11, the risk allele (rs2941471-A) was associated with higher expression of HNF4G in GTeX ($P = 0.038$, $\beta = 0.15$: T-statistic) and LTG ($P = 0.024$, $\beta = 0.28$: T-statistic) samples, but not in TCGA ($P = 0.80$, $\beta = -0.029$: T-statistic).

At 1p36.33/NOC2L, we analyzed the set of variants most likely to be functional (LR > 1:100, $n = 10$) for overlap with transcriptionally active chromatin and effects on predicted TF-binding sites. The most notable variant in this analysis was rs13303160 ($r^2 = 0.93$ with rs13303010) that overlaps open chromatin and prominent histone modification marks in ENCODE data (Fig. 2a,

Supplementary Table 10) and where the risk allele is predicted to strongly disrupt TF-binding motifs for SMARCC1 (also known as BAF155) and several AP-1 proteins (Fig. 2c, Supplementary Table 11). These analyses suggest that altered SMARCC1 or AP-1 binding at rs13303160 may lead to higher levels of NOC2L mRNA. An increase in NOC2L protein would be expected to result in lower levels of histone acetylation and repression of p53 and p63 transcriptional activity[24–26,59]. We also assessed differential expression of NOC2L (and additional genes at the five novel loci) in pancreatic tumors[44] and noted a 3.98-fold ($P = 9.69 \times 10^{-10}$: EdgeR, Exact test) increased expression in pancreatic cancer cell lines ($n = 9$) as compared with histologically normal pancreatic tissue samples ($n = 10$; Supplementary Table 12).

**Discussion**

This study demonstrates the power of large-scale collaborations in identifying new risk loci for pancreatic cancer, a deadly disease that presents challenges in accruing large sample sets for genetic studies. We herein add to the number of pancreatic cancer risk loci in or close to genes involved in MODY. As these genes also play roles in pancreatic development and acinar homeostasis, they may help explain underlying mechanisms at these loci. However, due to the low LD with BMI and T2D GWAS variants, the underlying mechanisms may differ between these epidemiologically and pathophysiologically associated conditions. We also describe potential functional underpinnings of risk loci, in particular for the locus on chromosome 1p36.33 in NOC2L, that require further detailed investigation.

**Methods**

**Study participants.** Participants were drawn from the Pancreatic Cancer Cohort Consortium (PanScan) and the Pancreatic Cancer Case-Control Consortium (PanC4) and individuals were included from 16 cohort and 13 case–control studies genotyped in four previous GWAS phases, namely PanScan I, PanScan II, PanScan III, and PanC4[12–14,16]. Samples from the PANDoRA case–control consortium were used for replication[23]. The details on cases (individuals with PDAC) and controls have been previously described[12–14,16].

All studies obtained informed consent from study participants and Institutional Review Board (IRB) approvals including IRB certifications permitting data sharing in accordance with the NIH Policy for Sharing of Data Obtained in NIH Supported or Conducted GWAS. The PanScan and PanC4 GWAS data are available through dbGAP (accession numbers phs000206.v5.p3 and phs000648.v1.p1, respectively).

**Genotyping, imputation, and association analysis.** Genotyping for PanScan was performed at the Cancer Genomics Research Laboratory (CGR) of the National Cancer Institute (NCI) of the National Institutes of Health (NIH) using the Illumina HumanHap series arrays (Illumina HumanHap550 Infinium II, Human 610-Quad) for PanScan I and II, respectively, and the Illumina Omni series arrays (OmniExpress, Omni1M, Omni2.5, and Omni5M) for PanScan III[12–14]. Genotyping for the PanC4 GWAS was performed at the Johns Hopkins Center for Inherited Disease Research (CIDR) using the Illumina HumanOmniExpressExome-8v1 array. Imputation was performed using the 1000 Genomes (1000 G) Phase 3, Release 1 reference data set[22] and IMPUTE2 (http://mathgen.stats.ox.ac.uk/impute/impute_v2.html)[60] as previously described[14,18]. Because of the large overlap of variants on genotyping arrays for PanScan I and II, these data sets were imputed and analyzed together. The PanScan III and PanC4 GWAS data sets were each imputed and analyzed separately. For quality control, variants were excluded based on (1) completion rate <90%; (2) MAF <0.01; and (3) low-quality imputation score (IMPUTE2 INFO score <0.3). After quality control, 11,381,182 SNPs genotyped or imputed in 5107 pancreatic cancer patients and 8845 controls of European ancestry were included in the analysis for PanScan I-III and 3933 cases and 3651 controls for PanC4 (Supplementary Table 1). The association analysis was performed using SNPTEST (http://mathgen.stats.ox.ac.uk/genetics_software/snptest/snptest.html)[61] based on probabilistic genotype values from IMPUTE2[60], with parallel covariate adjustments: study, geographical region, age, sex, and principal components (PCA) of population substructure as were used in PanScan[12–14] and study, age, sex, and PCA population substructure as were used in PanC4[16]. The score statistic of the log additive genetic association magnitude was used. Summary statistics from PanScan I and II, PanScan III, and PanC4 were used for a meta-analysis using the fixed-effects inverse-variance method based on $\beta$ estimates and SEs (http://csg.sph.umich.edu/abecasis/metal/). Heterogeneity was not observed for the SNPs identified as GWAS significant or suggestive in the combined study ($P_{heterogeneity} \geq 0.32$). IMPUTE2 information scores were above 90% for SNPs ($P < 1 \times 10^{-6}$), except for rs13303010 in the PanScan I+II data (INFO = 0.42;

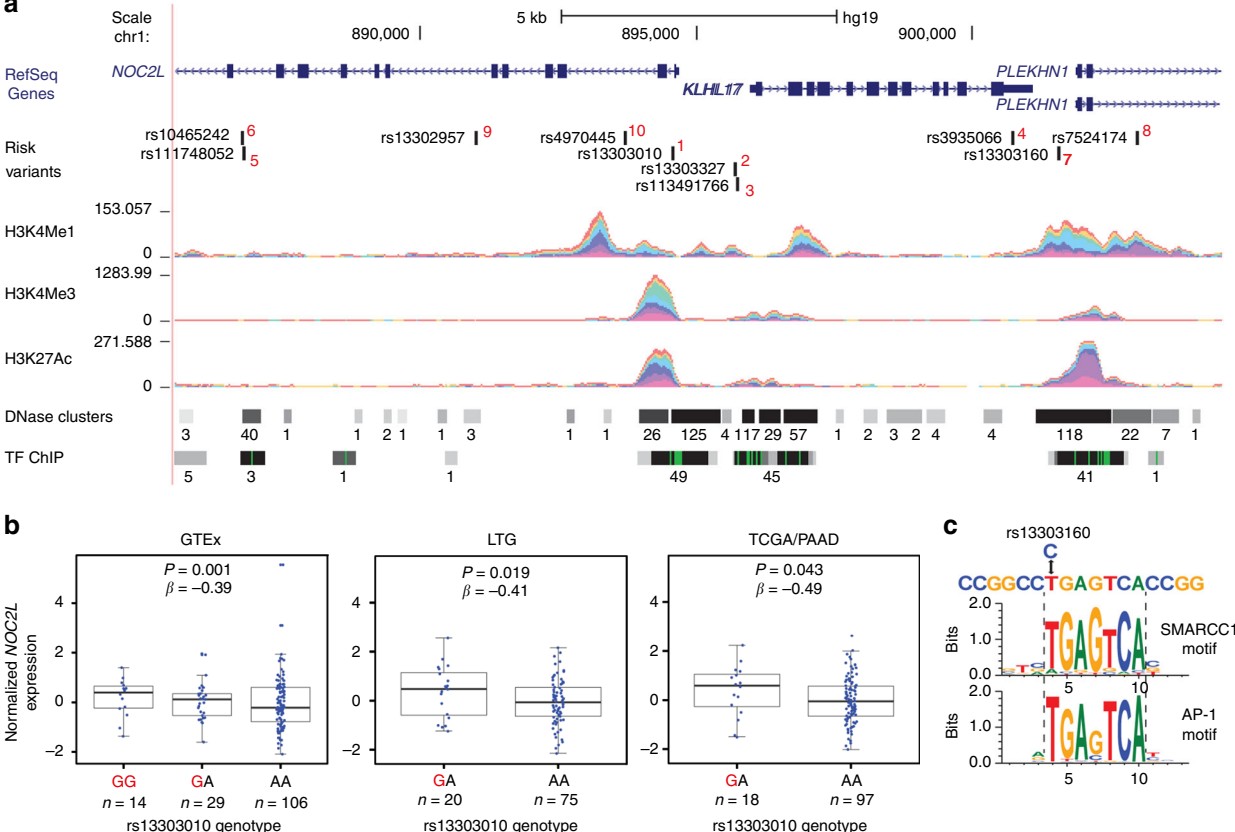

**Fig. 2** Functional analysis of the 1p36.33 risk locus. **a** The set of most likely functional variants at 1p36.33 and their *P* value rank (1–10, in red) is shown as well as overlapping RefSeq genes on chr1: 885,555-904,522 (NCBI GRCh37/Hg19). ENCODE data for histone modification marks (H3K4me1, H3K4me3, H3K27Ac) are indicated by colored density plots. Open chromatin (DNase hypersensitivity regions, DNase clusters) and binding of transcription factors (TF ChIP) are indicated by horizontal bars. The numbers next to each bar indicate the number of cell lines with DNase clusters, or the number of different transcription factors bound across all tested cell lines. The panel is adapted from the UCSC Genome Browser. **b** Expression QTLs in histologically normal autopsy-derived pancreatic tissues (*n* = 149) from the GTEx consortium (GTEx), the Laboratory of Translational Genomics histologically normal adjacent-to-tumor pancreatic tissue set (LTG, *n* = 95), and the TCGA pancreatic cancer tissue set (TCGA/PAAD, *n* = 115). Normalized *NOC2L* expression is shown on the *y* axis and genotypes at the marker SNP at 1p36.33 on the *x* axis. Risk-increasing alleles are marked in red. Note that no samples in the LTG and TCGA/PAAD sets were of the minor homozygous risk genotype (GG). The box-and-whisker plots show the median (horizontal middle line within each box), interquartile range (top and bottom horizontal lines of each box), and 1.5 times the IQR (whiskers). **c** Analysis of the effects of 1p36.33 variants on transcription factor motifs for rs13303160 (*r*² = 0.93 with rs13303010 in 1000 G EUR). The risk allele (C) at this marker alters predicted DNA-binding motifs for SMARCC1 and AP-1 proteins

Table 1). The estimated inflation of the test statistic, *λ*, was 1.002 for PanScan I+II, 1.051 for PanScan III, and 1.024 for PanC4[62].

A Polygenic Risk Score (PRS) was constructed for each individual by summing the number of risk alleles carried for all established pancreatic cancer risk loci identified by GWAS, weighted by their estimated effect size. Individuals were grouped by percentiles, and the association of the PRS (as percentile groupings) with pancreatic cancer was estimated using logistic regression.

DEPICT analysis was used to prioritize causal genes at currently known pancreatic cancer risk loci[12–16] identify gene sets enriched across risk loci, and tissues in which genes at risk loci are highly expressed[57]. No genes or SNP–gene pairs were significant at false discovery rate (FDR) < 0.05 (data not shown) but significant tissue enrichment is shown in Supplementary Table 7. Additional gene set enrichment analysis was performed using GeneCodis3 (http://genecodis.cnb. csic.es/). Genes (*n* = 65) were located in the currently known pancreatic cancer risk loci identified in subjects of European descent (for genes located +/−100 kb from the most significant SNP at the 22 risk loci) based on KEGG, Gene Ontology (Biological Process and Molecular Function) annotations using GeneCodis3 with reporting of FDR-corrected hypergeometric *P* values (Supplementary Table 8)[56].

**Replication**. Ten promising signals (*P* < 10⁻⁶) were selected for replication in samples from the PANDoRA consortium[23]. Genotyping was performed by custom TaqMan genotyping assays (Applied Biosystems) at the German Cancer Research Center (DKFZ) in Heidelberg, Germany, for 2770 pancreatic cancer patients and 5178 controls, of which 2737 cases and 4753 controls had complete age and clinical data and did not overlap with other study individuals. Duplicate quality-control samples (*n* = 607 pairs) showed 99.48% genotype concordance. SNP quality metrics

were performed for each SNP by plate; plates with <80% genotype completion rates were dropped from the analysis. Individuals were excluded if they were missing data on two or more SNPs after excluding SNPs on plates with low genotype completion rates. The association analysis for PANDoRA was adjusted for age and study in the same manner as previously described[14–16]. Heterogeneity between studies was assessed using the Cochran's Q-test. Association analysis was also performed for the set of variants previously replicated in PANDoRA as part of the PanScan III[14] and PanC4[16] GWAS studies (Supplementary Table 3).

Using SequenceLDhot, recombination hotspots for association plots were generated as previously described[12–14]. Recombination hotspot inference was performed using the 1000 G CEU samples (*n* = 99). The LD heatmap was prepared using the 1000 G Phase 3 CEU data, and the snp.plotter R software package[63].

**eQTL analysis**. The publicly available GTEx data[64] (http://www.gtexportal.org/; version 6) were used to assess eQTLs in pancreatic tissue samples (*n* = 149). RefSeq genes located within +/−500 kb of the marker SNP for each GWAS significant locus were assessed for *cis*-eQTL effects. Nominally significant eQTLs from this analysis (*P* < 0.05) were then taken forward to further analysis in two additional pancreatic tissue sample sets[58]: (1) the LTG and (2) The Cancer Genome Atlas (TCGA) pancreatic adenocarcinoma (PAAD) samples.

The LTG set included 95 histologically normal pancreatic tissue samples from participants of European ancestry collected at three participating sites: Mayo Clinic in Rochester, MN (45 samples, adjacent to tumor); Memorial Sloan Kettering Cancer Center in New York City, NY (34 samples, adjacent to tumor); and Penn State College of Medicine, Hershey, PA (16 samples, from tissue donors via the Gift of Life Donor Program, Philadelphia, PA) as previously described[58]. Samples were

confirmed to be non-tumorous with ≥80% epithelial component by histological review and macro-dissected when needed. The project was approved by the Institutional Review Board of each participating institution as well as the NIH.

In short, RNA (RIN >7.5) isolated from fresh frozen histologically normal pancreatic tissue samples (LTG samples) with the Ambion mirVana kit was poly-A-enriched and subjected to massively parallel paired-end sequencing (Illumina's HiSeq2000/TruSeq v3 sequencing). MapSplice was used to align reads and RSEM (v1.2.14) for gene expression quantification (TPM)[65,66] using the hg19/GRCh37-based UCSC "RefSeq" track for gene annotation. DNA for genotyping was isolated from blood (Mayo Clinic samples), histologically normal fresh frozen pancreatic tissue samples (Penn State samples), or histologically normal fresh frozen spleen or duodenum tissue samples (MSKCC samples) using the Gentra Puregene Tissue Kit (Qiagen). DNA samples were genotyped on the Illumina OmniExpress or Omni1M arrays at the CGR of the Division of Cancer Epidemiology and Genetics, NCI, NIH. After quality control, genotypes were imputed using the 1000 G (Phase 1, v3) imputation reference data set and IMPUTE 2[67]. Pre-imputation exclusion filters of Hardy Weinberg Equilibrium $P < 1 \times 10^{-6}$, minor allele frequency (MAF) <0.01, genotype missing rate >0.05, A/T and G/C pairs on ambiguous DNA strand (MAF > 0.45), and significantly different allele frequency between sample data and the 1000 G reference data ($P < 7 \times 10^{-8}$: Fisher's exact test) were used. Post-imputation variants (single-nucleotide variants (SNP) and small insertion-deletion (indel) polymorphisms) with MAF < 0.05 or imputation quality scores (INFO score) <0.5 were removed from the final analysis[58].

The second sample set included RNA-sequencing (RNA-seq) and genotype data from tumor-derived pancreatic samples obtained from TCGA PAAD data set by permission through the TCGA Data Access Committee. We excluded samples of non-European ancestry, with history of neo-adjuvant therapy prior to surgery, or with histological subtypes other than PDAC, leaving a total of 115 tumor samples for analysis[58]. TCGA mRNA-seq data (level 1 read data, generated using Illumina's HiSeq2000) for pancreatic cancer samples (TCGA PAAD) were processed in the same manner as the histologically normal LTG samples described above. Blood-derived DNA samples for TCGA PAAD samples were genotyped on Affymetrix 6.0 arrays and processed in the same manner as for the LTG samples[58].

The eQTL analysis was performed separately in histologically normal (LTG) and tumor-derived (TCGA PAAD) pancreatic samples using the Matrix eQTL (http://www.bios.unc.edu/research/genomic_software/Matrix_eQTL/) software package[68]. We tested associations between genotyped and imputed SNPs and the expression of genes evaluated by mRNA-sequencing after upper quantile normalization within samples and normal quantile transformation for each gene across samples by regressing the imputed dosage of the minor allele for each variant against normalized gene expression values[68]. Linear models were adjusted for age, sex, study, and the top five principal components (PCs) each for genotypes and gene expression to account for possible measured or hidden confounders[58]. The T-statistics from the linear regression is reported. For the tumor samples, we further adjusted for tumor stage and sequence-based tumor purity as per information provided by TCGA.

### Bioinformatic analysis of functional potential.
Variants at the new risk loci were assessed for potential functionality by examining their location in open (DNase Hypersensitivity Regions, DHS) and active chromatin (as per promoter and enhancer histone modification marks) in the ENCODE data. For this, we used HaploReg 4.1 (http://www.broadinstitute.org/mammals/haploreg/haploreg.php) and the UCSC Genome Browser (http://genome.ucsc.edu/). Variants correlated with the most significant variant at each locus at $r^2 > 0.7$ in 1000 G EUR populations were included, except for 1p36.33/NOC2L where $P$ value LR > 1:100 was used.

Candidate functional variants at 1p36.33 were selected by comparing the likelihood of each variant from the association analysis with the likelihood of the most significant variant. Ten variants had likelihood ratios, LR > 1:100 relative to the most significant SNP: rs13303010 ($P$ value rank 1), rs13303327 (rank 2), rs113491766 (rank 3), rs3935066 (rank 4), rs111748052 (rank 5), rs10465242 (rank 6), rs13303160 (rank 7), rs7524174 (rank 8), rs13302957 (rank 9), and rs4970445 (rank 10). They were all highly correlated with rs13303010 ($r^2 = 0.52-1.00$, 1000 G EUR data). These 10 variants were considered the set of variants most likely to contain the functional variant(s) at 1p36.33. Possible allelic effects of these top 10 variants on TF-binding motifs were determined using PrEdict Regulatory Functional Effect of SNPs by Approximate $P$ value Estimation (PERFECTOS-APE; http://opera.autosome.ru/perfectosape/) analysis that determines the probability of a TF motif (using position weight matrices, from HOCOMOCO-10, JASPAR, HT-SELEX, SwissRegulon, and HOMER databases) in the DNA sequence overlapping each variant. The fold change in probability of there being a TFBS present for each allele of a variant is then calculated[69]. Two dbSNP variants at 1p36.33 with the same bp location, rs111748052 (−/ATTTT) and rs10465241 (C/T), may be two independent variants (as indicated in dbSNP), or a single tri-allelic marker (alleles are shown as C/CATTTT/T in 1000 G). As PERFECTOS-APE does not analyze indel variants, we analyzed the two indels among the 10 potential functional variants, rs111748052 and rs113491766, using a different program, sequence TF Affinity Prediction. This program calculates the total affinity of a sequence for a TF (as given by TRANSFAC and JASPAR databases) on the basis of a biophysical model of the binding energies between a TF and DNA[70]. The probability for a given TFBS for each variant of the indel was then compared as in PERFECTOS-APE to determine the fold change effect of the indel on the presence of the TFBS.

### Gene expression analysis.
Gene expression was assessed for genes that are closest to the reported variants at chromosomes 1p36.33: *NOC2L*, *KLHL17*, and *PLEKHN1*; 7p12: *TNS3*; 8q21.11: *HNF4G*; 17q12: *HNF1B*; 18q21.32: *GRP*, as well as two additional genes at 1p36.33 exhibiting nominally significant eQTLs in GTEx (1p36.33/*SAMD11/DVL1*). We assessed differential expression of these genes in pancreatic tumor samples (PDAC, $n = 8$), histologically normal (non-malignant) pancreatic tissue samples ($n = 10$), and pancreatic cell lines ($n = 9$) by RNA-seq as described previously[44]. We compared gene expression in tumors (T) and cell lines (C) to histologically normal pancreatic tissue samples (N) by EdgeR analysis. $P$ values for differential expression in tumor vs. normal (TvN) and Cell lines vs. normal (CvN) represent an exact statistic using the normalized pseudo-counts and tagwise dispersion estimates per gene.

### Data availability.
The PanScan and PanC4 genome-wide association data that support the findings of this study are available through dbGAP (accession numbers phs000206.v5.p3 and phs000648.v1.p1, respectively).

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

## Acknowledgements

This work was supported by RO1 CA154823, the Lustgarten Foundation, and federal funds from the NCI, US NIH under contract number HHSN261200800001E. The content of this publication does not necessarily reflect the views or policies of the US Department of Health and Human Services, and mention of trade names, commercial products, or organizations does not imply endorsement by the US government. Genotyping Services were provided by the CIDR and the NCI's CGR. CIDR is fully funded through a federal contract from the NIH to the Johns Hopkins University, contract number HHSN268201100011I. The IARC/Central Europe study was supported by a grant from the US NCI at the NIH (R03 CA123546-02) and grants from the Ministry of Health of the Czech Republic (NR 9029-4/2006, NR9422-3, NR9998-3, and MH CZ-DRO-MMCI 00209805). The work at Johns Hopkins University was supported by the NCI Grants P50CA062924 and R01CA97075. Additional support was provided by, Susan Wojcicki, and Dennis Troper, and the Sol Goldman Pancreas Cancer Research Center. The Mayo Clinic Biospecimen Resource for Pancreas Research study is supported by the Mayo Clinic SPORE in Pancreatic Cancer (P50 CA102701). The Memorial Sloan Kettering Cancer Center Pancreatic Tumor Registry is supported by P30CA008748, the Geoffrey Beene Foundation, the Arnold and Arlene Goldstein Family, Foundation, and the Society of MSKCC. The PACIFIC Study was supported by RO1CA102765, Kaiser Permanente, and Group Health Cooperative. The Queensland Pancreatic Cancer Study was supported by a grant from the National Health and Medical Research Council of Australia (NHMRC; Grant number 442302). R.E.N. is supported by a NHMRC Senior Research Fellowship (#1060183). The UCSF pancreas study was supported by NIH-NCI grants (R01CA1009767, R01CA109767-S1, and R0CA059706) and the Joan Rombauer Pancreatic Cancer Fund. Collection of cancer incidence data was supported by the California Department of Public Health as part of the statewide cancer reporting program; the NCI's SEER Program under contract HSN261201000140C awarded to CPIC; and the CDC's National Program of Cancer Registries, under agreement

#U58DP003862-01 awarded to the California Department of Public Health. The Yale (CT) pancreas cancer study is supported by NCI at the U.S. NIH, grant 5R01CA098870. The cooperation of 30 Connecticut hospitals, including Stamford Hospital, in allowing patient access is gratefully acknowledged. The Connecticut Pancreas Cancer Study was approved by the State of Connecticut Department of Public Health Human Investigation Committee. Certain data used in that study were obtained from the Connecticut Tumor Registry in the Connecticut Department of Public Health. The authors assume full responsibility for analyses and interpretation of these data. Studies included in PAN-DoRA were partly funded by the Czech Science Foundation (No. P301/12/1734), the Internal Grant Agency of the Czech Ministry of Health (IGA NT 13 263); the Baden-Württemberg State Ministry of Research, Science and Arts (Professor H. Brenner), the Heidelberger EPZ-Pancobank (Professor M.W. Büchler and team: Professor T. Hackert, Dr. N. A. Giese, Dr. Ch. Tjaden, E. Soyka, M. Meinhardt; Heidelberger. Stiftung Chirurgie and BMBF grant 01GS08114), the BMBH (Professor P. Schirmacher; BMBF grant 01EY1101), the "5 × 1000" voluntary contribution of the Italian Government, the Italian Ministry of Health (RC1203GA57, RC1303GA53, RC1303GA54, and RC1303GA50), the Italian Association for Research on Cancer (Professor A. Scarpa; AIRC n. 12182), the Italian Ministry of Research (Professor A. Scarpa; FIRB - RBAP10AHJB), and the Italian FIMP-Ministry of Health (Professor A. Scarpa; 12 CUP_J33G13000210001), and by the National Institute for Health Research Liverpool Pancreas Biomedical Research Unit, UK. We would like to acknowledge the contribution of Dr. Frederike Dijk and Professor Oliver Busch (Academic Medical Center, Amsterdam, the Netherlands). Assistance with genotype data quality control was provided by Cecelia Laurie and Cathy Laurie at the University of Washington Genetic Analysis Center. The American Cancer Society (ACS) funds the creation, maintenance, and updating of the Cancer Prevention Study II cohort. Cancer incidence data for CLUE were provided by the Maryland Cancer Registry, Center for Cancer Surveillance and Control, Department of Health and Mental Hygiene, 201 W. Preston Street, Room 400, Baltimore, MD 21201, http://phpa.dhmh.maryland.gov/cancer, 410-767-4055. We acknowledge the State of Maryland, the Maryland Cigarette Restitution Fund, and the National Program of Cancer Registries of the Centers for Disease Control and Prevention for the funds that support the collection and availability of the cancer registry data. We thank all the CLUE participants. The Melbourne Collaborative Cohort Study (MCCS) recruitment was funded by VicHealth and Cancer Council Victoria. The MCCS was further supported by Australian NHMRC grants 209057 and 396414 and by the infrastructure provided by Cancer Council Victoria. Cases and their vital status were ascertained through the Victorian Cancer Registry and the Australian Institute of Health and Welfare, including the National Death Index and the Australian Cancer Database. The NYU study (AZJ and AAA) was funded by NIH R01 CA098661, UM1 CA182934 and center grants P30 CA016087 and P30 ES000260. The PANKRAS II Study in Spain was supported by research grants from Instituto de Salud Carlos III-FEDER, Spain: Fondo de Investigaciones Sanitarias (FIS; #PI13/00082 and #PI15/01573) and Red Temática de Investigación Cooperativa en Cáncer, Spain (#RD12/0036/0050); and European Cooperation in Science and Technology (COST Action #BM1204: EU_Pancreas), Ministerio de Ciencia y Tecnología (CICYT SAF 2000-0097), Fondo de Investigación Sanitaria (95/0017), Madrid, Spain; Generalitat de Catalunya (CIRIT—SGR); "Red temática de investigación cooperativa de centros en Cáncer" (C03/10), "Red temática de investigación cooperativa de centros en Epidemiología y salud pública" (C03/09), and CIBER de Epidemiología (CIBERESP), Madrid. The Physicians' Health Study was supported by research grants CA-097193, CA-34944, CA-40360, HL-26490, and HL-34595 from the NIH, Bethesda, MD, USA. The Women's Health Study was supported by research grants CA-047988, HL-043851, HL-080467, and HL-099355 from the NIH, Bethesda, MD, USA. Health Professionals Follow-up Study is supported by NIH grant UM1 CA167552 from the NCI, Bethesda, MD, USA. Nurses' Health Study is supported by NIH grants UM1 CA186107, P01 CA87969, and R01 CA49449 from the NCI, Bethesda, MD, USA. Additional support from the Hale Center for Pancreatic Cancer Research, U01 CA21017 from the NCI, Bethesda, MD, USA, and the United States Department of Defense CA130288, Lustgarten Foundation, Pancreatic Cancer Action Network, Noble Effort Fund, Peter R. Leavitt Family Fund, Wexler Family Fund, and Promises for Purple to B.M. Wolpin is acknowledged. The WHI program is funded by the National Heart, Lung, and Blood Institute, NIH, U.S. Department of Health and Human Services through contracts HHSN268201600018C, HHSN268201600001C, HHSN268201600002C, HHSN268201600003C, and HHSN268201600004C. The authors thank the WHI investigators and staff for their dedication, and the study participants for making the program possible. A full listing of WHI investigators can be found at http://www.whi.org/researchers/Documents%20%20Write%20a%20Paper/WHI%20Investigator%20Long%20List.pdf. We thank Laurie Burdett, Aurelie Vogt, Belynda

Hicks, Amy Hutchinson, Meredith Yeager, and other staff at the NCI's Division of Epidemiology and Genetics (DECG) CGR for GWAS genotyping. We also thank Bao Tran, Jyoti Shetty, and other members of the NCI Center for Cancer Research (CCR) Sequencing Facility for sequencing RNA from histologically normal pancreatic tissue samples (LTG samples). This study utilized the high-performance computational capabilities of the Biowulf Linux cluster at the NIH, Bethesda, MD, USA (http://biowulf.nih.gov). The Genotype-Tissue Expression (GTEx) Project was supported by the Common Fund of the Office of the Director of the NIH, and by NCI, NHGRI, NHLBI, NIDA, NIMH, and NINDS. The data used for the analyses described in this manuscript were obtained from the pancreatic tissue data from the GTEx Portal on 05/04/17. The results published here are in part based upon data generated by The Cancer Genome Atlas (TCGA) managed by the NCI and NHGRI. Information about TCGA can be found at http://cancergenome.nih.gov/. We acknowledge the clinical contributors that provided PDAC samples and the data producers of RNA-seq and GWAS genotype data from TCGA Research Network. The data set used for the analyses described in this manuscript was obtained by formal permission through the TCGA Data Access Committee (DAC).

## Author contributions

A.P.K., R.S.S., L.T.A., B.M.W., G.M.P., and H.A.R. organized and designed the study. A.P.K., L.T.A., F.C., J.W.H., A.J., J.Z., and M.B. organized and supervised the genotyping of samples. E.J.C., E.M., O.O., M.Z., A.Blackford. L.T.A., F.C., P.K., J.W.H., A.J., J.Z., F. Chen, and A.P.K. designed and conducted the statistical analysis. A.P.K. and L.T.A. drafted the first version of the manuscript. A.P.K., B.M.W., H.A.R., R.Z.S.-S., F.C., D.A., G.A., A.A.A., A.B., W.R.B., L.B.-F., S.I.B., A.Borgida, P.M.B., L.B., P.B., H.B., B.B.-d.-M., J.B., D.C., G.C., G.C., G.M.C., K.G.C., C.C.C., S.C., M.C., F.D., E.J.D., L.F., C.F., N.F., S.G., J.M.M.G., M.Gazouli, G.G.G., E.G., M.G., G.E.G., P.J.G., T.H., C.H., P.H., M.H., P.H., K.J. H., J.H., I.H., E.A.H., R.H., R.J.H., E.J.J., K.J., V.J., R.K., K.-T.K., E.A.K., M.K., C.K., M.H. K., J.K., R.J.K., D.L., S.L., R.T.L., I.-M.L., L.L., L.Lu., N.M., A.M., S.M., R.L.M., B.M.-D., R. E.N., J.P.N., A.L.O., I.O., C.P., A.V.P., U.P., R.P., M.P., F.X.R., N.R., G.S., H.D.S., G.S., X.-O.S., D.S., J.P.S., P.S., M.S., R.T.-W., F.T., M.D.T., G.S.T., S.K.V., Y.V., K.V., P.V., J.W.-W., Z.W., N.W., E.W., H.Y., K.Y., A.Z.-J., W.Z., P.K., D.L., S.C., O.O., G.M.P., and L.T.A. conducted the epidemiological studies and contributed samples to the GWAS, expression and/or follow-up genotyping. All authors contributed to the writing of the manuscript.

## Additional information

**Competing Interests:** The authors declare no competing financial interests.

Alison P. Klein[1,2], Brian M. Wolpin[3], Harvey A. Risch[4], Rachael Z. Stolzenberg-Solomon[5], Evelina Mocci[1], Mingfeng Zhang[6], Federico Canzian[7], Erica J. Childs[1], Jason W. Hoskins[6], Ashley Jermusyk[6], Jun Zhong[6], Fei Chen[1], Demetrius Albanes[5], Gabriella Andreotti[5], Alan A. Arslan[8,9,10], Ana Babic[3], William R. Bamlet[11], Laura Beane-Freeman[5], Sonja I. Berndt[5], Amanda Blackford[1], Michael Borges[2], Ayelet Borgida[12],

Paige M. Bracci[13], Lauren Brais[3], Paul Brennan[14], Hermann Brenner[15,16,17], Bas Bueno-de-Mesquita[18,19,20,21], Julie Buring[22,23], Daniele Campa[24], Gabriele Capurso[25], Giulia Martina Cavestro[26], Kari G. Chaffee[11], Charles C. Chung[5,27], Sean Cleary[12], Michelle Cotterchio[28,29], Frederike Dijk[30], Eric J. Duell[31], Lenka Foretova[32], Charles Fuchs[33], Niccola Funel[34], Steven Gallinger[12], J. Michael M. Gaziano[35,36], Maria Gazouli[37], Graham G. Giles[38,39,40], Edward Giovannucci[3], Michael Goggins[2], Gary E. Goodman[41], Phyllis J. Goodman[42], Thilo Hackert[43], Christopher Haiman[44], Patricia Hartge[5], Manal Hasan[45], Peter Hegyi[46], Kathy J. Helzlsouer[47], Joseph Herman[48], Ivana Holcatova[49], Elizabeth A. Holly[13], Robert Hoover[5], Rayjean J. Hung[12], Eric J. Jacobs[50], Krzysztof Jamroziak[51], Vladimir Janout[52,53], Rudolf Kaaks[54], Kay-Tee Khaw[55], Eric A. Klein[56], Manolis Kogevinas[57,58,59,60], Charles Kooperberg[41], Matthew H. Kulke[3], Juozas Kupcinskas[61], Robert J. Kurtz[62], Daniel Laheru[1], Stefano Landi[24], Rita T. Lawlor[63], I.-Min Lee[22,64], Loic LeMarchand[65], Lingeng Lu[4], Núria Malats[66,67], Andrea Mambrini[68], Satu Mannisto[69], Roger L. Milne[38,39], Beatrice Mohelníková-Duchoňová[70], Rachel E. Neale[71], John P. Neoptolemos[72], Ann L. Oberg[11], Sara H. Olson[73], Irene Orlow[73], Claudio Pasquali[74], Alpa V. Patel[50], Ulrike Peters[41], Raffaele Pezzilli[75], Miquel Porta[58,59], Francisco X. Real[67,76,77], Nathaniel Rothman[5], Ghislaine Scelo[14], Howard D. Sesso[22,23], Gianluca Severi[38,39,78], Xiao-Ou Shu[79], Debra Silverman[5], Jill P. Smith[80], Pavel Soucek[81], Malin Sund[82], Renata Talar-Wojnarowska[83], Francesca Tavano[84], Mark D. Thornquist[41], Geoffrey S. Tobias[5], Stephen K. Van Den Eeden[85], Yogesh Vashist[86], Kala Visvanathan[87], Pavel Vodicka[88], Jean Wactawski-Wende[89], Zhaoming Wang[90], Nicolas Wentzensen[5], Emily White[41,91], Herbert Yu[65], Kai Yu[5], Anne Zeleniuch-Jacquotte[9,92], Wei Zheng[79], Peter Kraft[23,93], Donghui Li[94], Stephen Chanock[5], Ofure Obazee[7], Gloria M. Petersen[11] & Laufey T. Amundadottir[6]

[1]Department of Oncology, Sidney Kimmel Comprehensive Cancer Center, Johns Hopkins School of Medicine, Baltimore, MD 21231, USA. [2]Department of Pathology, Sol Goldman Pancreatic Cancer Research Center, Johns Hopkins School of Medicine, Baltimore, MD 21287, USA. [3]Department of Medical Oncology, Dana-Farber Cancer Institute, Boston, MA 02215, USA. [4]Department of Chronic Disease Epidemiology, Yale School of Public Health, New Haven, CT 06520, USA. [5]Division of Cancer Epidemiology and Genetics, National Cancer Institute, National Institutes of Health, Bethesda, MD 20892, USA. [6]Laboratory of Translational Genomics, Division of Cancer Epidemiology and Genetics, National Cancer Institute, National Institutes of Health, Bethesda, MD 20892, USA. [7]Genomic Epidemiology Group, German Cancer Research Center (DKFZ), 69120 Heidelberg Germany. [8]Department of Obstetrics and Gynecology, New York University School of Medicine, New York, NY 10016, USA. [9]Department of Population Health, New York University School of Medicine, New York, NY 10016, USA. [10]Department of Environmental Medicine, New York University School of Medicine, New York, NY 10016, USA. [11]Department of Health Sciences Research, Mayo Clinic College of Medicine, Rochester, MN 55905, USA. [12]Lunenfeld-Tanenbaum Research Institute of Mount Sinai Hospital, Toronto, Ontario M5G 1×5, Canada. [13]Department of Epidemiology and Biostatistics, University of California, San Francisco, San Francisco, CA 94158, USA. [14]International Agency for Research on Cancer (IARC), 69372 Lyon France. [15]Division of Clinical Epidemiology and Aging Research, German Cancer Research Center (DKFZ), 69120 Heidelberg Germany. [16]Division of Preventive Oncology, German Cancer Research Center (DKFZ), 69120 Heidelberg Germany. [17]National Center for Tumor Diseases (NCT), 69120 Heidelberg Germany. [18]Department for Determinants of Chronic Diseases (DCD), National Institute for Public Health and the Environment (RIVM), 3720 BA Bilthoven The Netherlands. [19]Department of Gastroenterology and Hepatology, University Medical Centre, 3584 CX Utrecht The Netherlands. [20]Department of Epidemiology and Biostatistics, School of Public Health, Imperial College London, London SW7 2AZ, UK. [21]Department of Social and Preventive Medicine, Faculty of Medicine, University of Malaya, 50603 Kuala Lumpur Malaysia. [22]Division of Preventive Medicine, Brigham and Women's Hospital, Boston, MA 02215, USA. [23]Department of Epidemiology, Harvard T.H. Chan School of Public Health, Boston, MA 02115, USA. [24]Department of Biology, University of Pisa, 56126 Pisa Italy. [25]Digestive and Liver Disease Unit, 'Sapienza' University of Rome, 00185 Rome Italy. [26]Gastroenterology and Gastrointestinal Endoscopy Unit, Vita-Salute San Raffaele University, IRCCS San Raffaele Scientific Institute, 20132 Milan Italy. [27]Cancer Genomics Research Laboratory, National Cancer Institute, Division of Cancer Epidemiology and Genetics, Leidos Biomedical Research Inc., Frederick National Laboratory for Cancer Research, Frederick, MD 21702, USA. [28]Cancer Care Ontario, University of Toronto, Toronto, Ontario M5G 2L7, Canada. [29]Dalla Lana School of Public Health, University of Toronto, Toronto, Ontario M5T 3M7, Canada. [30]Department of Pathology, Academic Medical Center, University of Amsterdam, 1007 MB Amsterdam The Netherlands. [31]Unit of Nutrition and Cancer, Cancer Epidemiology Research Program, Bellvitge Biomedical Research Institute (IDIBELL), Catalan Institute of Oncology (ICO), Barcelona 08908, Spain. [32]Department of Cancer Epidemiology and Genetics, Masaryk Memorial Cancer Institute, 65653 Brno Czech Republic. [33]Yale Cancer Center, New Haven, CT 06510, USA. [34]Department of Translational Research and The New Technologies in Medicine and Surgery, University of Pisa, 56126 Pisa Italy. [35]Division of Aging, Brigham and Women's Hospital, Boston, MA 02115, USA. [36]Boston VA Healthcare System, Boston, MA 02132, USA. [37]Department of Basic Medical Sciences, Laboratory of Biology, Medical School, National and Kapodistrian University of Athens, 106 79 Athens Greece. [38]Cancer Epidemiology and Intelligence Division, Cancer Council Victoria, Melbourne, VIC 3004, Australia. [39]Centre for Epidemiology and Biostatistics, Melbourne School of Population and Global Health, The University of Melbourne, Parkville, VIC 3010, Australia. [40]Department of Epidemiology and Preventive Medicine, Monash University, Melbourne, VIC 3004, Australia. [41]Division of Public Health Sciences, Fred Hutchinson Cancer Research Center, Seattle, WA 98109, USA. [42]SWOG Statistical Center, Fred Hutchinson Cancer Research Center, Seattle, WA 98109, USA. [43]Department of General Surgery, University Hospital Heidelberg, 69120 Heidelberg Germany. [44]Department of Preventive Medicine, Keck School of Medicine, University of Southern California, Los Angeles, CA 90032, USA. [45]Department of Epidemiology, University of Texas MD Anderson Cancer Center, Houston, TX 77230, USA. [46]First Department of Medicine,

University of Szeged, 6725 Szeged Hungary. [47]Division of Cancer Control and Population Sciences, National Cancer Institute, National Institutes of Health, Bethesda, MD 20892, USA. [48]Department of Radiation Oncology, Sidney Kimmel Comprehensive Cancer Center, Johns Hopkins School of Medicine, Baltimore, MD 21231, USA. [49]Institute of Public Health and Preventive Medicine, Charles University, 2nd Faculty of Medicine, 150 06 Prague 5 Czech Republic. [50]Epidemiology Research Program, American Cancer Society, Atlanta, GA 30303, USA. [51]Department of Hematology, Institute of Hematology and Transfusion Medicine, 02-776 Warsaw Poland. [52]Department of Epidemiology and Public Health, Faculty of Medicine, University of Ostrava, 701 03 Ostrava Czech Republic. [53]Faculty of Medicine, University of Olomouc, 771 47 Olomouc Czech Republic. [54]Division of Cancer Epidemiology, German Cancer Research Center (DKFZ), 69120 Heidelberg Germany. [55]School of Clinical Medicine, University of Cambridge, Cambridge CB2 0SP, UK. [56]Glickman Urological and Kidney Institute, Cleveland Clinic, Cleveland, OH 44195, USA. [57]ISGlobal, Centre for Research in Environmental Epidemiology (CREAL), 08003 Barcelona Spain. [58]CIBER Epidemiología y Salud Pública (CIBERESP), 08003 Barcelona Spain. [59]Hospital del Mar Institute of Medical Research (IMIM), Universitat Autònoma de Barcelona, 08003 Barcelona Spain. [60]Universitat Pompeu Fabra (UPF), 08002 Barcelona Spain. [61]Department of Gastroenterology, Lithuanian University of Health Sciences, 44307 Kaunas Lithuania. [62]Department of Medicine, Memorial Sloan Kettering Cancer Center, New York, NY 10065, USA. [63]ARC-NET: Centre for Applied Research on Cancer, University and Hospital Trust of Verona, 37134 Verona Italy. [64]Department of Epidemiology, Harvard School of Public Health, Boston, MA 02115, USA. [65]Cancer Epidemiology Program, University of Hawaii Cancer Center, Honolulu, HI 96813, USA. [66]Genetic and Molecular Epidemiology Group, Spanish National Cancer Research Center (CNIO), 28029 Madrid Spain. [67]CIBERONC, 28029 Madrid Spain. [68]Oncology Department, ASL1 Massa Carrara, Carrara 54033, Italy. [69]Department of Public Health Solutions, National Institute for Health and Welfare, 00271 Helsinki Finland. [70]Department of Oncology, Faculty of Medicine and Dentistry, Palacky University Olomouc and University Hospital, 775 20 Olomouc Czech Republic. [71]Population Health Department, QIMR Berghofer Medical Research Institute, Brisbane 4029, Australia. [72]Department of General Surgery, University of Heidelburg, Heidelberg Germany. [73]Department of Epidemiology and Biostatistics, Memorial Sloan Kettering Cancer Center, New York, NY 10065, USA. [74]Department of Surgery, Oncology and Gastroenterology (DiSCOG), University of Padua, 35124 Padua Italy. [75]Pancreas Unit, Department of Digestive Diseases and Internal Medicine, Sant'Orsola-Malpighi Hospital, 40138 Bologna Italy. [76]Epithelial Carcinogenesis Group, Spanish National Cancer Research Centre-CNIO, 28029 Madrid Spain. [77]Departament de Ciències Experimentals i de la Salut, Universitat Pompeu Fabra, 08002 Barcelona Spain. [78]Centre de Recherche en Épidémiologie et Santé des Populations (CESP, Inserm U1018), Facultés de Medicine, Université Paris-Saclay, UPS, UVSQ, Gustave Roussy, 94800 Villejuif France. [79]Division of Epidemiology, Department of Medicine, Vanderbilt Epidemiology Center, Vanderbilt-Ingram Cancer Center, Vanderbilt University School of Medicine, Nashville, TN 37232, USA. [80]Department of Medicine, Georgetown University, Washington 20057, USA. [81]Laboratory for Pharmacogenomics, Biomedical Center, Faculty of Medicine in Pilsen, Charles University, 323 00 Pilsen Czech Republic. [82]Department of Surgical and Perioperative Sciences, Umeå University, 901 85 Umeå Sweden. [83]Department of Digestive Tract Diseases, Medical University of Łodz, 90-647 Łodz Poland. [84]Division of Gastroenterology and Research Laboratory, IRCCS Scientific Institute and Regional General Hospital "Casa Sollievo della Sofferenza", 71013 San Giovanni Rotondo, FG, Italy. [85]Division of Research, Kaiser Permanente Northern California, Oakland, CA 94612, USA. [86]Department of General, Visceral and Thoracic Surgery, University Hamburg-Eppendorf, 20246 Hamburg Germany. [87]Department of Epidemiology, Johns Hopkins Bloomberg School of Public Health, Baltimore, MD 21205, USA. [88]Department of Molecular Biology of Cancer, Institute of Experimental Medicine, Academy of Sciences of the Czech Republic, 142 20 Prague 4 Czech Republic. [89]Department of Epidemiology and Environmental Health, University at Buffalo, Buffalo, NY 14214, USA. [90]Department of Computational Biology, St. Jude Children's Research Hospital, Memphis, TN 38105, USA. [91]Department of Epidemiology, University of Washington, Seattle, WA 98195, USA. [92]Perlmutter Cancer Center, New York University School of Medicine, New York, NY 10016, USA. [93]Department of Biostatistics, Harvard School of Public Health, Boston, MA 02115, USA. [94]Department of Gastrointestinal Medical Oncology, University of Texas MD Anderson Cancer Center, Houston, TX 77030, USA. Alison P. Klein, Brian M. Wolpin, Harvey A. Risch and Rachael Z. Stolzenberg-Solomon contributed equally to this work. Donghui Li, Stephen Chanock, Ofure Obazee, Gloria M. Petersen and Laufey T. Amundadottir jointly supervised this work.

