## [Peer Review File · Nature Communications]

Reviewers' comments:

Reviewer #1 Expert in pancreatic cancer genetics:

This is the largest GWAS performed to date for pancreatic cancer by the leaders in the field with robust application of well established methods.

The only comment I would make is that increasing numbers will invariably identify more risk alleles of diminishing effect size. Based on the diversity of individual events within genes and loci in familial pancreatic cancer studies and that seen in somatic genomes; it would be beneficial to begin to understand how these events may tie into functional groups. This could be done in the context of somatic genomes as well as for susceptibility alleles. We likely miss many germ line events that contribute to carcinogenesis, and may confer therapeutic vulnerabilities, but are not associated with increased risk compared to the mean population risk.

Reviewer #2 Expert in cancer GWAS:

Klein and colleagues describe a meta-analysis of pancreatic cancer GWAS, where they identify novel loci. The methods are straightforward and the paper is clearly written. This manuscript adds significantly to the field of pancreatic cancer genetic epidemiology. I have a few suggestions of edits.

The 7p12 SNP failed to replicate in PANDoRA and is not genome-wide significant in the whole dataset. This should be mentioned in the abstract, especially since it was included among the ten loci that were followed up through replication. Do the authors believe that the lack of replication was due to lack of power or something else? (minor point: The text (page 9) says that the PANDoRA replication p-value for rs73328514 is 0.38 but Table 1 states $p=0.31$)

It would be informative with more information about the underlying study population. Even though the authors refer to previous papers, it is tedious for the reader to go back to each and one of them to learn more about the studies included and the general distribution of gender. How many cases were incident vs prevalent?

It would be interesting to see a look-up of pancreatic cancer SNPs identified in Asian ancestry populations in this European ancestry based GWAS. Even though these loci might not reach genome-wide significance in Europeans, do the authors observe any indication of a signal for the 8 regions identified in Asians?

It would be interesting to see a more general description of the findings beyond description of individual loci including

How much of the variation in pancreatic cancer do GWAS-identified variants explain?

Did the author assess the burden of calculating multiple variants associated with pancreatic cancer (e.g. compare the top 10% and 5% with the median based on adding up risk variants)?

Did the authors attempt any enrichment or pathway-based analysis to find common themes among GWAS-identified regions (e.g DEPICT or other analyses)?

Did the associations for these new loci differ by gender, age or other relevant covariates? The authors should be able to assess this using data from PANDoRA, where they have the raw genotype data.

Minor comments

Include qq and manhattan plots for the meta-analysis

When discussing the HNF1B region, (top on page 10) the authors first mention that their top SNP is correlated with a prostate cancer SNP ($r^2=0.59$) but in the next sentence go on to say that SNPs in this region associated with prostate cancer (and endometrial and testicular cancer) are not correlated with the new pancreatic cancer SNPs. Are the authors discussing two different independent prostate cancer SNPs? Please clarify.

Comma signs in the rsid for chr7 SNP in Table 1 should be fixed

Suppl Table 8 is mistakenly labeled Suppl Table 6

In Gene expression analysis (Online methods), the authors state that they assess five genes that were closest to the top SNPs on Chr1p36.33 but then only mention 3: NOC2L, KLHL17 and PLEKHN1. They later mention SAMD11 and DVL1 but it is not clear if those are included in the five genes? (I do not see the DVL1 gene in Figure 1a, is it located outside the region?)

When discussing the eQTL results for rs13303010 and NOC2L (page 11), the authors state that "the risk allele at rs13303010 was associated with higher NOC2L expression in all three datasets (GTEx: $P=0.01$, $\beta= -0.39$; LTG: $P=0.019$, $\beta= -0.41$; TCGA: $P=0.043$, $\beta= -0.49$)". For clarity, I suggest explicitly stating which allele is the risk allele and update the beta estimates for eQTL analysis accordingly to reflect their previous statement that gene expression is increased (positive beta estimates rather than negative).

Row 6 on page 11: I think that rs1330160 should read rs13303160

We appreciate the comments from both reviewers and have performed the additional analyses suggested and updated the manuscript. In the rebuttal below text added to the manuscript is shown in underlined text.

Reviewers' comments:

Reviewer #1 Expert in pancreatic cancer genetics:

This is the largest GWAS performed to date for pancreatic cancer by the leaders in the field with robust application of well-established methods.

The only comment I would make is that increasing numbers will invariably identify more risk alleles of diminishing effect size. Based on the diversity of individual events within genes and loci in familial pancreatic cancer studies and that seen in somatic genomes; it would be beneficial to begin to understand how these events may tie into functional groups. This could be done in the context of somatic genomes as well as for susceptibility alleles. We likely miss many germ line events that contribute to carcinogenesis, and may confer therapeutic vulnerabilities, but are not associated with increased risk compared to the mean population risk.

We agree there is much still to be learned about the germline changes that predispose individuals to pancreatic cancer and how these relate to somatic events and carcinogenesis. Our manuscript represents the first step in this process by identifying five new pancreatic cancer susceptibility loci, in addition to the already known risk loci from our prior work, and the first steps in the process of understanding their functional underpinnings using eQTL analysis in pancreatic tissues, bioinformatic analysis of transcription factor binding sites affected by risk SNPs, and differential expression analysis of candidate functional genes in normal and tumor derived pancreatic samples. However, further study is needed to understand in detail the mechanisms by which the relatively common germline variants at these loci contribute to the development of pancreatic cancer. The expression data we present for *NOC2L* provides some evidence that genetic variation at this locus may be associated with increased *NOC2L* levels. However, further functional studies are still needed.

We have also used pathway analysis for genes in currently known pancreatic cancer risk loci (+/- 100kb of the marker SNP) and identified *MODY* (Maturity Onset Diabetes of the Young) and the most enriched pathway with genes such as *HNF1A* (chr12q24), *HNF1B* (chr17q12), *HNF4G* (chr8q21), *PDX1* (chr13q12) and *NR5A2* (chr1q32) highlighted (see response to reviewer #2 below). These genes are not only linked to *MODY* but also play important roles in pancreatic development and the maintenance of acinar homeostasis that may represent important determinants of pancreatic carcinogenesis. However, this hypothesis will need to be tested by functional wet-lab approaches and possibly also animal models. We are also very interested in the potential links between germline and somatic events that influence pancreatic cancer and plan to look at this in more detail in future studies.

Reviewer #2 Expert in cancer GWAS:

Klein and colleagues describe a meta-analysis of pancreatic cancer GWAS, where they identify novel loci. The methods are straightforward and the paper is clearly written. This

manuscript adds significantly to the field of pancreatic cancer genetic epidemiology. I have a few suggestions of edits.

The 7p12 SNP failed to replicate in PANDoRA and is not genome-wide significant in the whole dataset. This should be mentioned in the abstract, especially since it was included among the ten loci that were followed up through replication. Do the authors believe that the lack of replication was due to lack of power or something else? (minor point: The text (page 9) says that the PANDoRA replication p-value for rs73328514 is 0.38 but Table 1 states $p=0.31$)

We have revised the manuscript abstract to state that rs73328514 was not significantly associated in the PANDoRA population. However, given the OR is in the same direction, (OR=0.94 in PANDoRA vs OR=0.83 in PanScan and PanC4) we feel this is most likely an issue of power.

Text added in abstract: The locus at 7p12 was not significantly associated with pancreatic cancer risk in PANDoRA (rs78417682, OR=0.94 95% CI 0.83-1.06, $P=0.31$).

We would like to thank the reviewer for calling our attention to the incorrect P -value in the text. This has been corrected.

It would be informative with more information about the underlying study population. Even though the authors refer to previous papers, it is tedious for the reader to go back to each and one of them to learn more about the studies included and the general distribution of gender. How many cases were incident vs prevalent?

This information has been added to Supplemental Table 1.

It would be interesting to see a look-up of pancreatic cancer SNPs identified in Asian ancestry populations in this European ancestry based GWAS. Even though these loci might not reach genome-wide significance in Europeans, do the authors observe any indication of a signal for the 8 regions identified in Asians?

We have now added Supplementary Table 6 that provides the results for the 8 regions identified in Asians in our meta-analysis and the following text in the revised manuscript:

We also assessed 8 pancreatic cancer risk loci identified in Chinese and Japanese individuals in our data and noted one nominally significant locus in the combined PanScan and PanC4 results (6p25.3, rs9502893, OR=0.94, $P=0.009$) (Supplemental Table 6).

It would be interesting to see a more general description of the findings beyond description of individual loci including. How much of the variation in pancreatic cancer do GWAS-identified variants explain?

We agree this would be of interest and examined several approaches to determine heritability, however due to the limitations of the data we have in-hand we felt these analyses did not add to our previously published results. While we were able to estimate heritability using LDSC for each population, given that only summary statistics were shared between the PanScan and PanC4 groups, we were unable to estimate the locus specific heritability. We therefore explored heritability using other methods including

HESS, however, our correspondence with the developers indicated this method requires sample sizes a least 50,000 for accurate estimation of locus specific heritability.

Did the author assess the burden of calculating multiple variants associated with pancreatic cancer (e.g. compare the top 10% and 5% with the median based on adding up risk variants)?

We have added these analyses. The manuscript now states: “We further estimated a polygenetic risk score (PRS) for pancreatic cancer using the 22 independent genome-wide significant risk SNPs in the Caucasian population. The odds ratio for pancreatic cancer among individuals above the 90th percentile the risk distribution was 2.20 (95% CI 1.83-2.65) compared to those with a polygenetic risk score (PRS) in the 40-60th percentile (**Supplemental Table 5**).”

Did the authors attempt any enrichment or pathway-based analysis to find common themes among GWAS-identified regions (e.g DEPICT or other analyses)?

We have performed DEPICT analysis for the risk loci identified in the current analysis as well as those we have previously published in PanScan and PanC4 GWAS. While we do not see significant enrichment of specific pathways (FDR<0.05), we do see enrichment of MESH terms relating to the digestive system (14/18 MESH terms) including multiple gastrointestinal organs. The pancreas MESH term has a nominal P value of 0.013 but is not significant after FDR correction.

We also performed classical pathway enrichment analysis for genes within each risk locus with a boundary of 100 kb up and downstream of the most significant SNP. Again, we included both our current risk loci as well as those that we have previously published in individuals of European ancestry. The most significant enrichment was seen for the terms “Maturity onset diabetes of the young” (KEGG, $P=5.5 \times 10^{-9}$), “Sequence-specific DNA binding transcription factor activity” (GO Molecular function, $P=3.1 \times 10^{-4}$) and “Cellular response to UV” (GO Biological Process, $P=4.2 \times 10^{-4}$).

We have now included results from this analysis as Supplemental Tables 7 and 8 in the manuscript and added methods as well as text in the main manuscript text:

Pathway enrichment analysis for genes in currently known pancreatic cancer risk loci was performed using gene set enrichment analysis and Data-driven Expression Prioritized Integration for Complex Traits (DEPICT). The most significant enrichment was seen for the terms “Maturity onset diabetes of the young” (KEGG, $P=5.5 \times 10^{-9}$), “Sequence-specific DNA binding transcription factor activity” (GO Molecular Function, $P=3.1 \times 10^{-4}$), “Cellular response to UV” (GO Biological Process, $P=4.2 \times 10^{-4}$) as well as multiple gastrointestinal tissues (DEPICT, $P=5.16 \times 10^{-5}$ -0.004) (**Supplementary Tables 7 and 8**).

Did the associations for these new loci differ by gender, age or other relevant covariates? The authors should be able to assess this using data from PANDORA, where they have the raw genotype data.

We have completed these analyses and did not see significant differences by gender or age. We added the following sentence to the manuscript text:

For each of these loci, the effect sizes were consistent when stratified by age and gender.

Minor comments

Include qq and manhattan plots for the meta-analysis:

We have now added the meta-analysis QQ plot in Supplemental Figure 1.

When discussing the HNF1B region, (top on page 10) the authors first mention that their top SNP is correlated with a prostate cancer SNP ($r\text{-sq}=0.59$) but in the next sentence go on to say that SNPs in this region associated with prostate cancer (and endometrial and testicular cancer) are not correlated with the new pancreatic cancer SNPs. Are the authors discussing two different independent prostate cancer SNPs? Please clarify.

Yes, we are discussing two independent risk signals in the *HNF1B* gene region that associate with risk of prostate cancer. One of those SNPs is rs4794758 ($r^2=0.59$ with the PDAC tag SNP at 17q12/*HNF1B*, rs4430796) that we list in the manuscript. The other locus is marked by rs4430796 ($r^2=0.0007$ with our PDAC tag SNP).

We have now included the additional rs numbers in the second sentence on page 9 to clarify:

Although additional variants in this region have been associated with other cancers including prostate, endometrial and testicular cancers, they do not appear to mark the same signal (rs4430796, rs11263763, rs7501939, respectively, r^2 with rs4795218 ≤ 0.005 in 1000G EUR).

Comma signs in the rsid for chr7 SNP in Table 1 should be fixed

This has been corrected

Suppl Table 8 is mistakenly labeled Suppl Table 6

This has been corrected

In Gene expression analysis (Online methods), the authors state that they assess five genes that were closest to the top SNPs on Chr1p36.33 but then only mention 3: NOC2L, KLHL17 and PLEKHN1. They later mention SAMD11 and DVL1 but it is not clear if those are included in the five genes? (I do not see the DVL1 gene in Figure 1a, is it located outside the region?)

We are sorry for the confusion and have changed the methods accordingly (see underline and strikethrough text):

Changed from: "Gene expression was assessed for five genes that are closest to the reported variants at chromosomes 1p36.33: *NOC2L*, *KLHL17*, and *PLEKHN1*; 7p12: *TNS3*; 8q21.11: *HNF4G*; 17q12: *HNF1B*; 18q21.32: *GRP*, and those with nominally significant eQTLs in GTEx (1p36.33/*SAMD11/DVL1*)."

Changed to: “Gene expression was assessed for ~~five~~ genes that are closest to the reported variants at chromosomes 1p36.33: *NOC2L*, *KLHL17*, and *PLEKHN1*; 7p12: *TNS3*; 8q21.11: *HNF4G*; 17q12: *HNF1B*; 18q21.32: *GRP*, as well as two additional genes at chr1p36.33 exhibiting nominally significant eQTLs in GTEx (1p36.33/*SAMD11*/*DVL1*).”

DVL1 is located at chr1:1,270,658-1,284,492 or approximately 400 kb downstream of *NOC2L* and rs13303010. This gene was included in the eQTL analysis (as it is located within the +/- 500 kb boundary of the most significant SNP tagging this locus). It is correct that it is not included in Figure 1a that focuses on the pancreatic cancer GWAS signal (chr1:790,465-1,015,817).

When discussing the eQTL results for rs13303010 and NOC2L (page 11), the authors state that “the risk allele at rs13303010 was associated with higher NOC2L expression in all three datasets (GTEx: P=0.01, β = -0.39; LTG: P=0.019, β = -0.41; TCGA: P=0.043, β = -0.49)”. For clarity, I suggest explicitly stating which allele is the risk allele and update the beta estimates for eQTL analysis accordingly to reflect their previous statement that gene expression is increased (positive beta estimates rather than negative).

We agree that clarifying the direction of eQTL effect for the risk allele of rs13303010 is important and have changed the sentence accordingly:

Changed from: “the risk allele at rs13303010 was associated with higher *NOC2L* expression in all three datasets (GTEx: P=0.01, β = -0.39; LTG: P=0.019, β = -0.41; TCGA: P=0.043, β = -0.49)”.

Changed to: “the risk increasing allele at rs13303010 (G) was associated with higher *NOC2L* expression in all three datasets (GTEx: P=0.01, β =0.39; LTG: P=0.019, β =0.41; TCGA: P=0.043, β =0.49)”.

We have also changed the direction of effect accordingly in Table 2, Supplemental Figure 2 and Supplemental Table 9 to indicate the direction of effect for the risk increasing allele at each locus.

Row 6 on page 11: I think that rs1330160 should read rs13303160.

This has been corrected

REVIEWERS' COMMENTS:

Reviewer #1 (Remarks to the Author):

The authors have adequately addressed my comments.

Reviewer #2 (Remarks to the Author):

The authors have addressed my comments. I have no further comments.